# Endophytic Fungi Regulate *HbNHX1* Expression and Ion Balance in *Hordeum bogdanii* under Alkaline Stress

**DOI:** 10.3390/jof9030331

**Published:** 2023-03-07

**Authors:** Feng Long, Meng-Fei Hu, Sheng Chen, Gen-Sheng Bao, Han Dan, Shui-Hong Chen

**Affiliations:** 1College of Life Science and Technology, Tarim University, Alar 843300, China; 2Qinghai Academy of Animal and Veterinary Sciences, Xinning 810016, China

**Keywords:** alkali stress, endophytic fungi, *Hordeum bogdanii*, cations, anion, *HbNHX1* expression

## Abstract

Plants cope with abiotic stress in several ways, including by collaborating with microorganisms. *Epichloë,* an endophytic fungus, has been shown to improve plant tolerance to extreme external environments. *Hordeum bogdanii* is a known salt-tolerant plant with the potential to improve alkaline lands. *NHX1* plays a key role in the transport of ions in the cell and is overexpressed in plants with increased salt tolerance. However, the expression levels of *HbNHX1* in *Epichloë* endophytic fungal symbionts in *H. bogdanii* have not been elucidated. We used *Hordeum bogdanii* (E+) with the endophytic fungi *Epichloë bromicola* and *H. bogdanii* (E−) without the endophytic fungi and compared the differences in the ion content and *HbNHX1* expression between the shoots and roots of E+ and E− plants under alkaline stress. The absorption capacity of both K^+^ and Na^+^ of *H. bogdanii* with endophytic fungi was higher than that without endophytic fungi. In the absence of alkaline stress, endophytic fungi significantly reduced the Cl^−^ content in the host *H. bogdanii*. Alkaline stress reduced SO_4_^2−^ content in *H. bogdanii*; however, compared with E−, endophytic fungi increased the content of SO_4_^2−^ in E+ plants. With an increase in the alkaline concentration, the expression of *HbNHX1* in the roots of *H. bogdanii* with endophytic fungus exhibited an upward trend, whereas the expression in the shoots exhibited a downward trend first and then an upward trend. Under 100 mmol·L^−1^ mixed alkaline stress, the expression of *HbNHX1* in E+ was significantly higher than that in E−, indicating that endophytic fungi could increase the Na^+^ region in vacuoles. The external environment affects the regulation of endophytic fungi in *H. bogdanii* and that endophytic fungi can play a key role in soil salinization. Therefore, the findings of this study will provide technical support and a theoretical basis for better utilization of endophytic fungi from *H. bogdanii* in saline land improvement.

## 1. Introduction

Soil salinization has severely endangered the environment and the production of economic crops in China [1]. High salt levels can lead to ion toxicity, osmotic stress, oxidative damage, and other secondary stresses in plants [2], making it necessary for the plants to maintain the balance between ions in their cells to ensure normal growth and development. A large area of the inland saline-alkaline land in China mainly contains three cations: Na^+^, K^+^, Mg^2+^, and four anions: CO_3_^2−^, HCO_3_^−^, Cl^−^, and SO_4_^2−^ [3]. Excess accumulation of Na^+^ and Cl^−^ will result in ion toxicity, thus reducing the absorption of essential nutrients such as calcium (Ca), potassium (K), phosphorus (P), and nitrogen (N) [4,5]. As soil salinization involves not only salt stress but also alkaline stress, the same stress factors apply to alkaline stress as they do to salt stress, which also involves high pH stress [6], which leads to a significant amount of metal ions to precipitate, inhibits the absorption of anions, and breaks the ion balance inside and outside cells [7]. Therefore, to slow down the accumulation of cations such as Na^+^ and K^+^ under alkaline stress, plants maintain the balance of anions and cations by accumulating inorganic anions such as Cl^−^, NO_3_^−^, and SO_4_^2−^.

The effects of alkaline stress on the transport of Na^+^ and K^+^ in plants have been extensively studied. Plants cope with a large amount of Na^+^ accumulation under saline-alkaline stress mainly in two ways: *SOS1* gene involved ion efflux and the *NHX1* gene participation in intracellular ion regionalization. *SOS1* is located in the plasma membrane, expressed in the roots of epidermal cells and wood parenchyma cells, and is mainly involved in the efflux of Na^+^ from the cytoplasm to the soil and the transport of Na^+^ to leaves through transpiration [1,8]. *NHX1* is located in the vacuole membrane and plays an important role in the transport of Na^+^ in plants by exchanging Na^+^ into vacuoles and H^+^ into the cytoplasm. This reduces the excessive accumulation of Na^+^ in the cytoplasm and maintains the osmotic potential in cells so that they can absorb water normally [9]. In Arabidopsis and rice, plants overexpressing *NHXs* can retain K^+^ in the cytoplasm under saline conditions, thereby improving salt tolerance [10,11]. Pérez-Martín et al. [12] found that high expression of *NHX1* in Arabidopsis leaves under saline and alkaline environment can enhance salt tolerance by promoting Na^+^ partition and osmotic regulation. Plants can improve their tolerance to Na^+^ by collaborating with microorganisms [13]. For example, in tomatoes and lettuce, inoculation of plant roots with fungal endophytes resulted in an improvement in biomass, photosynthetic capacity, and survival rate under salt stress. At the same time, it enhanced the expression of the *NHX1* gene and promoted Na^+^ regionalization [14]. However, many molecular studies have mainly focused on the interaction between root endophytic fungi and plants, whereas there are few studies on the interaction of endophytic fungi in plant stems and leaves.

The endophytic fungus of grass spends part or all its life cycle affecting the growth process of grass and does not cause obvious symptoms in plants [15]. Under saline-alkaline stress, infection with endophytic fungi of Gramineae can improve the tolerance and adaptability of host plants to the external environment [16,17]. Recent research on endophytic fungi of Gramineae mainly focused on the genus *Epichloë* of Clavicipitaceae, Ascomycota [18], which affects the resource allocation and mineral absorption capacity of host plants [19]. 

*H. bogdanii* is a salt-tolerant plant with strong ecological adaptability and competitive advantages. It has broad prospects for development as a forage grass and to improve the ecological environments of desert, saline, and alkaline lands [20]. Recently, studies on *H. bogdanii*-endophytic fungi symbionts have mainly focused on the classification and identification of endophytic fungi [21,22,23] and the effects of endophytic fungi on the growth and development [23,24], physiology, and biochemistry [25,26] of *H. bogdanii* under abiotic stress. However, *HbNHX1* expression and ion balance of *Epichloë* endophytic fungal symbionts in *H. bogdanii* under saline-alkaline stress have not yet been reported. 

In this study, we aimed to explore the regulation mechanism of endophytic fungi on the expression of the *HbNHX1* gene of *H. bogdanii* and the balance of anions and cations in plants under alkali concentration stress. To achieve this, we measured *HbNHX1* expression and Na^+^, K^+^, Cl^−^, SO_4_^2−^ and NO_3_^−^ content in the shoots and roots of E+ and E− *H. bogdanii* plants under alkaline stress, and analyzed the effects of endophytic fungi on *HbNHX1* gene expression and ion balance regulation of the host *H. bogdanii* under alkaline stress.

## 2. Materials and Methods

### 2.1. Plant Materials and Culture Conditions

The seeds of Wensu E+ and E− *H. bogdanii* had been previously treated in our laboratory [25]. *Epichloë bogdanii* was used in this study [22]. The seeds were then planted in an experimental field (80°76′ E, 41°58′ N, H 1514 m) in the experimental station of Tarim University. E+ and E− plump seeds were germinated in water until they grew needles and then transplanted into 4 L hydroponic boxes, 19 holes per pot, spaced at 1 seedling per hole for a total of 19. Each basin was connected to an oxygen-charging pump, the air was regularly exchanged every day, and 1/2 Hoagland’s solution was replaced every three days. The day/night temperature of the greenhouse was 28 ± 2 °C/15 ± 3 °C and the sunshine duration was approximately 12 h. Once *H. bogdanii* tillered 3–5 plants, aniline blue microscopy [27] was used to determine the presence of endophytic fungi in *H. bogdanii*.

### 2.2. Stress Treatment

In the mixed alkaline (Na_2_CO_3_:NaHCO_3_ = 1:1) treatment, the concentration gradients were set to 50 mmol·L^−1^ and 100 mmol·L^−1^. The experiment had a randomized complete block design with four replicates (hydroponic box) of E+ and E− seedlings. To avoid alkali shock, the alkali concentration was increased by 50 mmol·L^−1^ every 12 h until all treatments reached the predetermined concentration. A solution without mixed alkali was used as the control for the experiment. The mixed alkali solution with the corresponding concentration was replaced every 7 d, and the sample was taken at 21 d for subsequent ion content and *NHX1* gene expression measurements.

### 2.3. Determination of Ion Content

After 21 days of alkali stress treatment, the *H. bogdanii* plants were divided into shoots and roots for the determination of ion content. Some samples were placed in an oven at 105 °C for 30 min and then dried to a constant weight at 70 °C. Shoots and roots samples were ground and sieved, weighed to 0.3 g and 0.1 g, respectively, and added to 10 mL of triacid (HNO_3_:HClO_4_:H_2_SO_4_ = 8:1:1) for digestion, and diluted to 50 mL. Na^+^ and K^+^ contents were determined using a flame photometer (FP6431) [28]. The other set of samples was placed in a vacuum freeze-drying instrument and freeze-dried for 24 h. Then, the shoots and roots samples were ground, sieved, and weighed to 0.5 g and 0.3 g, respectively. These ground samples were added to 10 mL of deionized water, fully ground, centrifuged at 3000 rpm for 10 min, and the supernatant was collected; this process was repeated three times. The supernatant was combined with deionized water to a volume of 50 mL, filtered with an aqueous phase filter membrane, and the contents of Cl^−^, SO_4_^2−^, and NO_3_^−^ were determined by ion chromatography (Dionex Integration HPIC high-pressure ion chromatography series) [29]. Each processing set was repeated four times.

### 2.4. Cloning of HbNHX1 Gene

Total RNA was extracted using TransZol (Full Gold, Beijing, China) and reverse transcribed into cDNA (TRAN HiFiScript^®^ gDNA Removal cDNA Synthesis Kit, Beijing, China). The reference barley Na^+^/H^+^ antiporter *HvNHX1* gene full-length sequence, accession number AB089197.1, designed two pairs of primers (*HbNHX1*): primers and reaction conditions, as shown in Table 1. The *HbNHX1* gene was cloned using Nested PCR via Bio-Rad PCR instrument (Hercules, California, USA, T100). After electrophoretic separation, the target fragment was recovered and constructed into the pMD19-T vector, and inserted into *E. coli* DH5α, cultured in LB liquid medium for 5 h, and then coated on solid LB medium for overnight culture. A single colony was selected and placed in LB liquid medium for amplification and culturing for 6 h. Recombinant bacteria with positive PCR results were sequenced (Xi’an Qingke Biotechnology Co., Ltd., Beijing, China).

### 2.5. Expression Analysis of HbNHX1 Gene

The reagent for fluorescent quantitative PCR was TRAN PerfectStart ^®^ (Green qPCR SuperMix, Beijing, China), and the real-time fluorescence quantitative PCR instrument (CFX96 Touch) from BIO-RAD Company (Hercules, California, USA) was used. The internal reference actin primer sequence was extracted from Vinje [30]. The qPCR primer design was carried out using the full-length CDS region of the *HbNHX1* gene obtained in the previous step. The primers were synthesized by Xi’an Qingke Biotechnology Co. The relevant information is shown in Table 1. The 2^−ΔΔCT^ method was used for calculation, and each treatment was repeated four times.

### 2.6. Data Analysis

SPSS statistical software (IBMSSSStatistics26) was used for data analysis. The data in the figures are presented as the mean ± standard error. The effects of different alkali concentrations on the ion content of endophytic fungi in *H. bogdanii* were detected using double-factor ANOVA, and the effects of different alkali concentrations on ion content and *HbNHX1* gene expression in *H. bogdanii* were detected using single-factor ANOVA. An independent sample *t*-test was used to detect the influence of E+ and E− ion content on *HbNHX1* gene expression under the same alkali concentration. Statistical significance was defined at a 95% confidence level. The online software DeepTMHMM was used to predict the transmembrane region of the protein encoded by the *HbNHX1* gene of *H. bogdanii*. Using the Neighbor-Joining method, a phylogenetic analysis of the *NHX1* gene was carried out in MEGAX64 software.

## 3. Results

### 3.1. Effect of Alkali Stress on Na^+^, K^+^ Content and K^+^/Na^+^ Ratio in E+, E− H. bogdanii

After alkaline treatment, K^+^ content in the plants was significantly lower than that in the control group. With an increase in alkaline concentration, the K^+^ content decreased continuously, but it was higher in E+ than that in E− (Figure 1A,B), and the content of K^+^ at 50 mmol·L^−1^ E+ is significantly higher than that of E−. The K^+^ content of the E+ shoots was significantly higher than that of E− in the 100 mmol·L^−1^ mixed alkaline treatment, but there was no significant difference between E+ and E− in the roots. According to the two-factor ANOVA of variance, endophytic fungi significantly affected K^+^ content in the shoots and roots of *H. bogdanii* (Table 2).

Under no stress, Na^+^ content was lower in E+ than that in E− and reached a significant level in the roots (Figure 1C,D). It showed that the endophytic fungi could reduce the absorption of Na^+^ by the host in the absence of alkali stress. However, it is interesting that the content of Na^+^ in the shoots and roots of the plants after alkali treatment shows an upward trend, and the Na^+^ content of the E+ plants was significantly higher than that of the E− plants. The possible reason for this phenomenon is that endophytic fungi promote the localization of host plant Na^+^. According to the two-factor ANOVA of variance, endophytic fungi significantly affected Na^+^ content in the shoots and roots of *H. bogdanii* (Table 2).

The K^+^/Na^+^ ratio of E+ was significantly higher than that of E− in the control group. However, after 50 mmol·L^−1^ mixed alkali stress, the K^+^/Na^+^ ratio of E+ was significantly lower than that of E− in the shoots, whereas in the roots, the K^+^/Na^+^ ratio of E+ was higher than that of E−. In the 100 mmol·L^−1^ mixed alkali treatment, the K^+^/Na^+^ ratio of E+ was significantly higher than that of E− in the shoots, but there was no significant difference in the roots (Figure 1E,F). According to the two-factor ANOVA in Table 2, alkaline treatment and endophytic fungi significantly affected the K^+^/Na^+^ ratio in the shoots and roots of *H. bogdanii* and there was an interaction between them (*p* < 0.05).

### 3.2. Effect of Alkali Stress on Cl^−^, SO_4_^2−^ and NO_3_^−^ Content in E+, E− H. bogdanii

In the absence of alkaline stress, the Cl^−^ content in the shoots and roots of E+ plants was significantly lower than that of E− plants. However, in the 50 mmol·L^−1^ mixed alkaline treatment, the Cl^−^ content of E− shoots was significantly higher than that of E+ shoots, but the Cl^−^ content E− of the roots was significantly lower than that of E+ roots. There was no significant difference between the Cl^−^ content in E+ and E− shoots at 100 mmol·L^−1^ mixed alkaline treatment, but the Cl^−^ content in E+ roots was significantly lower than that of E− roots (Figure 2A,B). According to the two-factor ANOVA in Table 3, alkaline treatment and endophytic fungi significantly affected the Cl^−^ content in the shoots and roots of *H. bogdanii*, respectively. Thus, Endophytic fungi interacted with the alkaline stress (*p* < 0.05).

The content of SO_4_^2−^ in E+ shoots was significantly higher than that of E− shoots under alkaline-free and alkaline stress conditions (Figure 2C). The content of SO_4_^2−^ in the roots of both plants gradually decreased with an increase in the alkaline concentration, and there was no significant difference in the content of SO_4_^2−^ in E+ and E− plants (Figure 2D). According to the two-factor variance analysis in Table 3, alkaline stress significantly affected SO_4_^2−^ content in the shoots and roots of *H. bogdanii*; thus, endophytic fungi interacted with alkaline stress in the shoots of *H. bogdanii* (*p* < 0.05).

Alkaline stress had no significant effect on the NO_3_^−^ content in the shoots of E+ and E− plants. No significant difference was observed between the E+ and E− groups (Figure 2E). The NO_3_^−^ content in the roots of E− plants increased and then decreased with an increase in alkaline concentration, whereas in E+ plants, it decreased and then increased accordingly. The NO_3_^−^ content in E+ roots was significantly lower than that in E− in the 50 mmol·L^−1^ mixed alkaline treatment, but the NO_3_^−^ content in E+ plants was significantly higher than that in E− in the 100 mmol·L^−1^ mixed alkaline treatment (Figure 2F). According to the two-way analysis of variance in Table 3, alkaline stress and endophytic fungi significantly affected the NO_3_^−^ content in the roots of *H. bogdanii*, and there was an interaction between them (*p* < 0.05).

### 3.3. Cloning Results of HbNHX1 Gene in H. bogdanii

Using E− *H. bogdanii* cDNA as the template, nested PCR was conducted to obtain approximately 1700 bp bands (Figure 3). After connecting the T vector with the recovered target band, the total length of the target band was found to be 1708 bp, including 1617 bp in the CDS region, encoding 538 amino acids in Appendix A

### 3.4. Bioinformatics Analysis of HbNHX1 Gene in H. bogdanii

The physical and chemical properties of the HbNHX1 protein were analyzed using ProtParam online software. The predicted results revealed that the molecular weight of the HbNHX1 protein was 59.28 kD, the isoelectric point (PI) was 7.68, and the elemental composition was C_2746_H_4286_N_668_O_739_S_25_. In order to study whether HbNHX1 protein is a transmembrane protein, we used TMHMM software to analyze its structure. Topological analysis showed that the protein had 13 transmembrane domains and belonged to the transmembrane protein (Figure 4). Through homology comparison, the *HbNHX1* had the highest similarity (96.29%) with the wild barley *NHX1* gene (AB089197.1), belonging to the gramineous barley genus, while the similarity with wheat (MT550001.1) was 94.99%, and the similarity with monocotyledonous model plant rice (KY752545.1) was 86.98% (Figure 5). 

### 3.5. Effect of Alkali Stress on HbNHX1 Gene Expression Level in Shoots and Roots of H. bogdanii

After alkaline stress, the expression of the *HbNHX1* gene in E+ shoots decreased and then increased with the increase in alkaline concentration compared to the control group. Under different alkaline stress conditions, E− had higher expression of the *HbNHX1* gene under alkaline stress, with the highest level at 50 mmol·L^−1^ mixed alkaline treatment. There was no significant difference in *HbNHX1* expression between E+ and E− shoots in the non-mixed alkaline treatment; however, after inducing alkaline stress, there was a significant difference in the expression of *HbNHX1* between E+ and E− shoots. 

In the shoots, compared with the control, the expression of *HbNHX1* gene in E+ decreased and then increased with an increase in alkaline concentration. At the same time, compared with the control, the expression of *HbNHX1* increased in E− plants under different alkaline stress and reached a peak at 50 mmol·L^−1^ mixed alkali treatment. There was no significant difference in the expression of *HbNHX1* between E + and E− shoots in the 0 mmol·L^−1^ mixed alkaline treatment, but the expression of *HbNHX1* in E+ and E− shoots was significantly different after alkaline stress. The expression of the E+ *HbNHX1* was significantly lower than that of E− at 50 mmol·L^−1^ mixed alkali treatment, while the expression of the E+ *HbNHX1* gene was significantly higher than that of E− at 100 mmol·L^−1^ mixed alkali treatment (Figure 6A). The expression of *HbNHX1* in the roots changed. Alkali treatment significantly increased the expression of *HbNHX1* in the roots of E+ plants, whereas the change in E− plants was not significant. There was no significant difference in the content of *HbNHX1* between E+ and E− in 0 mmol·L^−1^ mixed alkaline treatment roots, but the expression of *HbNHX1* in 50 mmol·L^−1^ and 100 mmol·L^−1^ mixed alkaline treatment E+ plants was significantly higher than that in E− plants (Figure 6B) (*p* < 0.05).

## 4. Discussion

### 4.1. Effect of Alkali Stress on Na^+^, K^+^ Content, and K^+^/Na^+^ Ratio in H. bogdanii

Na^+^ and K^+^ have similar radii of hydrated ions, so it is difficult to distinguish them, leading to a competitive relationship between the absorption of Na^+^ and K^+^ by roots [5]. Excessive salt absorption by plants causes ion toxicity. Plants transport Na^+^ in cells to vacuoles under salt stress to reduce water loss. In most plants living in saline-alkaline environments, Na^+^ accumulates in vacuoles in large quantities and K^+^ absorption is inhibited [31]. To cope with the excessive accumulation of Na^+^ in the cytoplasm of leaves under salt stress, *Hordeum brevisubulatum* transported it to vacuoles for Na^+^ localization to reduce Na^+^ toxicity [32]. Many studies have shown that endophytic infection can improve abiotic and biological resistance of host plants [33,34,35]. However, there have also been reports showing no significant differences between plants with and without endophytic fungi [36]. Our results showed that endophytic fungi and alkaline stress significantly affected the content of Na^+^, K^+^, and K^+^/Na^+^ in plants. Under alkaline stress, the content of Na^+^ in the shoots and roots of E+ and E− *H. bogdanii* exhibited an increasing trend, which is similar to the results published by Song [37]. However, this study found that alkaline stress led to Na^+^ content in E+ being significantly higher than that in E-, which may be because endophytic fungi promote the accumulation of Na^+^ in vacuoles during alkaline stress, leading to an increase in the content of Na^+^. K^+^ is an important mineral nutrient that can protect plants from damage. We found that the K^+^ content in the shoots and roots of *H. bogdanii* decreased significantly under alkaline stress, indicating that alkaline stress inhibits the absorption of K^+^ by plants. In addition, the K^+^ content of E+ was significantly higher than that of E−, indicating that *Epichloë* endophytic fungi can increase the K^+^ content in host plants. K^+^ content was higher in the shoots than that in the roots under alkaline stress, which is consistent with the results of Chen et al. [38], indicating that plants will accelerate K^+^ upward transport when subjected to alkaline stress to alleviate Na^+^ toxicity and maintain ionic balance. Alkaline stress increased the plant Na^+^ content and continuously decreased the K^+^ content, leading to a decrease in the K^+^/Na^+^ ratio, which was consistent with the trend of K^+^/Na^+^ ratio decline in *Peucedanum littoralis* and *Hordeum vulgare* [39] after salt stress.

### 4.2. Effect of Alkali Stress on Cl^−^, SO_4_^2−^ and NO_3_^−^ Content in H. bogdanii

Alkaline stress forces a large amount of Na^+^ in plants to increase, leading to an ion imbalance [40,41]. Plants usually accumulate inorganic anions, such as Cl^−^ [42,43], SO_4_^2−^, and NO_3_^−^, or synthetic organic anions [44] to regulate ion imbalance. In this study, we found that when subjected to alkaline stress, Cl^−^ content was significantly higher than that of the other anions. Additionally, under alkaline stress, the Cl^−^ content of E+ was lower than that of E−, and the Cl^−^ content of E− plants decreased continuously but E+ increased, indicating that endophytic fungi can regulate the Cl^−^ content to regulate the ion balance in plants and reduce the ionic damage in host plants. These results were consistent with studies that reported similar trends in Cl^−^ content in citrus (*Carrizo citrange*) [45], spring wheat (*Triticum aestivum*), and winter barley (*Hordeum vulgare*) [46] caused by mycorrhiza. We also found that alkaline stress reduced the content of SO_4_^2−^ in *H. bogdanii*, which is consistent with the findings of Yang et al. [47]. The SO_4_^2−^ content of E+ was higher than that of E− when endophytic fungi infected *H. bogdanii*, indicating that endophytic fungi increased the accumulation of SO_4_^2−^ in host plants. In plants, the increase in Cl^−^ content in the roots is related to a decrease in NO_3_^−^ content [48]. High pH caused by alkali stress may inhibit the absorption of nitrate [49], thus weakening the positive role of NO_3_^−^ in maintaining plant alkaline tolerance [50]. In addition, we found that when subjected to mixed alkali concentrations of 50 mmol·L^−1^ and 100 mmol·L^−1^, the Cl^−^ content in the roots of E+ increased then decreased, and the content of NO_3_^−^ decreased and then increased, which also confirms that the increase in Cl^−^ content is related to the decrease in NO_3_^−^ content. After alkaline stress, the content of SO_4_^2−^ and NO_3_^−^ in the roots decreased. Alkaline stress significantly affected the content of NO_3_^−^ in E+ and E− roots. At 50 mmol·L^−1^, NO_3_^−^ content in E+ was significantly lower than that in E−, whereas, at 100 mmol·L^−1^, NO_3_^−^ content in E+ was significantly higher than that in E−, indicating that the presence of endophytic fungi can regulate the change in NO_3_^−^ content in host plants to maintain the ionic balance after alkaline stress.

### 4.3. Effect of Alkali Stress on HbNHX1 Expression in H. bogdanii Roots and Shoots

Plants maintain ionic stability by controlling the expression of ion transporters. For example, many plants can help reduce Na^+^ toxicity and affect the expression of Na^+^-related genes by symbiosis with fungi [51]. *NHX1* is a key gene involved in plant alkaline tolerance. Overexpression of *NHX1* has been shown to improve salt tolerance in many plants. The existence of beneficial fungus mycorrhizae affected the gene expression of host plants of Pakchoi (*Brassica campestris* ssp. *Chinensis*) under salt stress. Moreover, the expression level of *NHX1* was higher in root tissues in inoculated plants under both saline and non-saline stress [52]. In this study, it was found that the expression of *HbNHX1* was highest in the stems and leaves of E+ *H. bogdanii* treated with 100 mmol·L^−1^ alkaline, while the expression of *HbNHX1* was lower in the stems and leaves treated with 50 mmol·L^−1^, indicating that under high salt, the endophytic fungi promoted the host *H. bogdanii* to regionalize more Na^+^ in the vacuoles, promoting the upregulation of *HbNHX1*. Diao [53] found that *Suaeda salsa* inoculated with *Arbuscular mycorrhizal* downregulated the expression of *SsNHX1* in roots under 100 mmol·L^−1^ NaCl. However, this study found that under alkaline stress, *HbNHX1* expression increased in the roots of *H. bogdanii* with endophytic fungi, and the expression of *HbNHX1* in E+ was significantly higher than that in E− under alkaline stress. The results demonstrated that endophytic fungi increased the expression of *HbNHX1* in the roots of *H. bogdanii*, and that different fungi had different regulatory effects on plants.

## 5. Conclusions

The results showed that *Epichloë* endophytic fungi promoted the upregulation of *HbNHX1* in the host *H. bogdanii* in an alkaline environment and participated in the regulation of anion and cation transport in the host plant. This study found that in an alkaline environment, *Epichloë* endophytic fungi reduced Na^+^ toxicity in *H. bogdanii* by increasing *HbNHX1* gene expression, thereby regionalizing Na^+^ in the vacuole membrane and contributing to the increase in K^+^ and related anions to regulate the effect of alkaline stress. This conclusion provides basic data for the innovation of germplasm resources with endophytic fungi from *H. bogdanii*, provides a scientific basis and strong guarantee for the breeding of new varieties and the healthy and stable development of grassland animal husbandry, and improves the stress tolerance of new varieties in alkaline environments.

## Figures and Tables

**Figure 1 jof-09-00331-f001:**
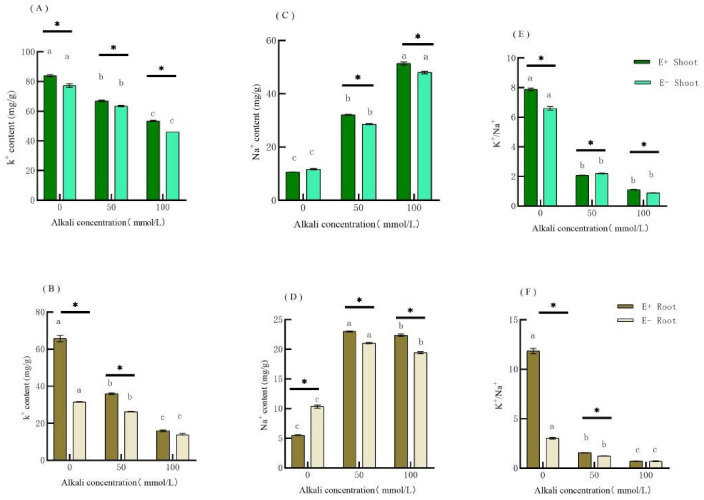
Effects of different alkali concentrations on K^+^ (**A**,**B**), Na^+^ (**C**,**D**) content and K^+^/Na^+^ (**E**,**F**) in E+ and E− *H. bogdanii* shoots and roots. Note: Among different groups, when comparing the same E+ or E− plant, different lowercase letters of the column mark indicate a significant difference (*p* < 0.05); in the same group E+ compared with E−, * indicates a significant difference (*p* < 0.05).

**Figure 2 jof-09-00331-f002:**
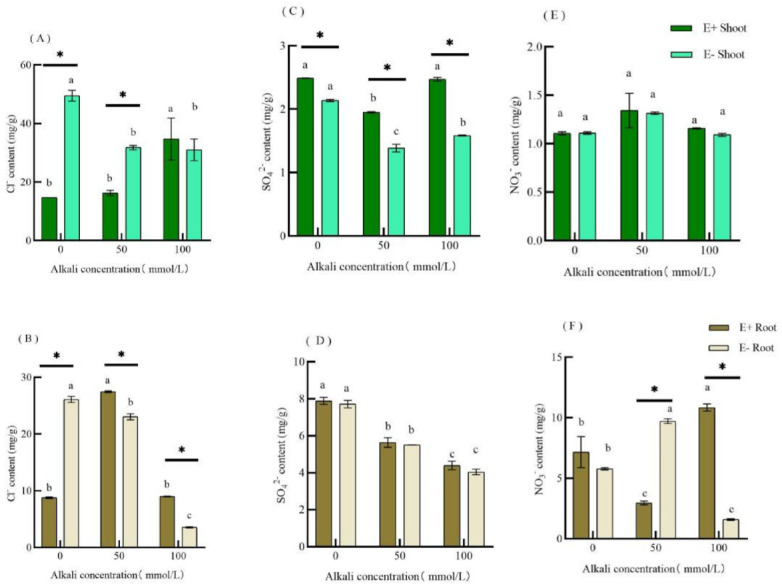
Effects of different alkali concentrations on Cl^−^ (**A**,**B**), SO_4_^2−^ (**C**,**D**) and NO_3_^−^ (**E**,**F**) contents in E+ and E− *H. bogdanii* shoots and roots.Note: Among different groups, when comparing the same E+ or E− plant, different lowercase letters of the column mark indicate a significant difference (*p* < 0.05); in the same group E+ compared with E−, * indicates a significant difference (*p* < 0.05).

**Figure 3 jof-09-00331-f003:**
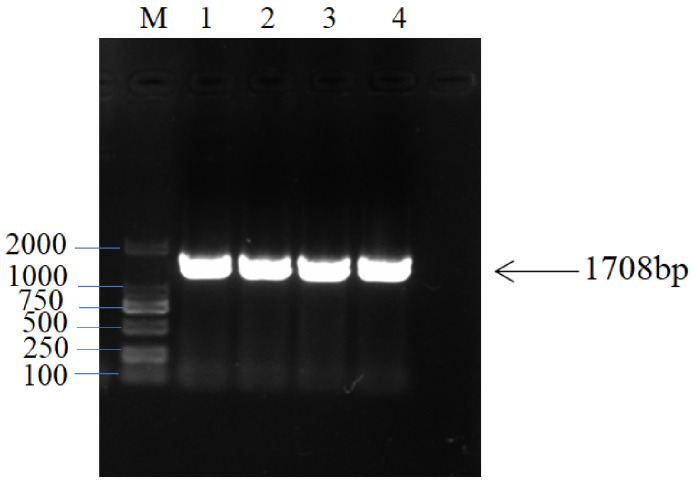
*HbNHX1* Clones. Note: M = Marker (2000 bp; 1). 2, 3, and 4 are *HbNHX1* genes.

**Figure 4 jof-09-00331-f004:**
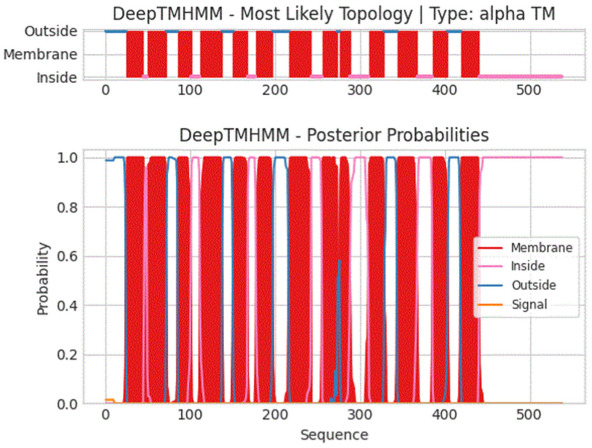
Structural analysis of HbNHX1 protein.The topological structure of the HbNHX1 protein was analyzed using TMHMM software (https://dtu.biolib.com/DeepTMHMM, accessed on 3 March 2020). The hydrophobic sequence with 13 transmembrane regions is shown in the large rectangle.

**Figure 5 jof-09-00331-f005:**
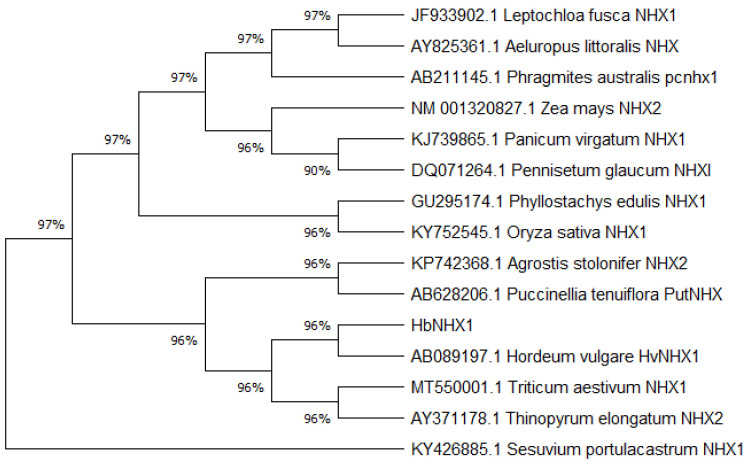
Evolutionary tree analysis of plant *NHX1* gene.

**Figure 6 jof-09-00331-f006:**
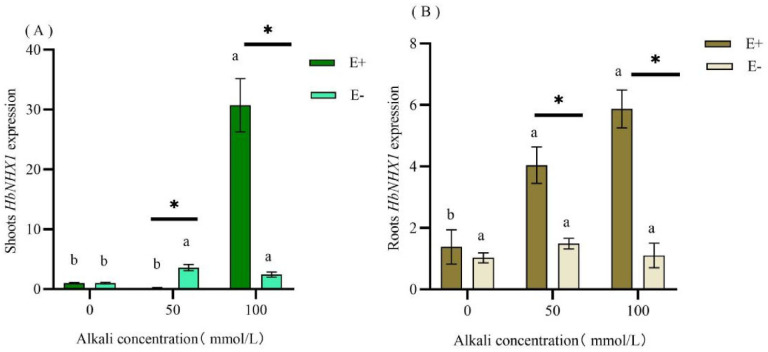
Effect of alkali stress on *HbNHX1* gene expression levels in E+, E− shoots (**A**) and roots (**B**) of *H. bogdanii. *Note: Among different groups, when comparing the same E+ or E− plant, different lowercase letters of the column mark indicate a significant difference (*p* < 0.05); in the same group E+ compared with E−, * indicates a significant difference (*p* < 0.05).

**Table 1 jof-09-00331-t001:** Gene primer sequence.

Gene	Primer Sequence	Reaction Procedure
*HvHNX1*	1F: GAAAAGAATAGAGGAGAATCCCGAC	The PCR conditions were 95 °C for 3 min, then 30 cycles of 94 °C for 30 s, 56.2 °C for 30 s, 72 °C for 90 s and a final extension at 72 °C for 10 min.
	1R: ACCATTACACCAATCCACTAGAAAG	
	2F: ATGGCGTTCGAAGTGGTGGC	
	2R: TCATGCCACGATTACGTTTGGA	
*HbNHX1*	F: CTGTGTCGATCGCTCTTGCT	The qPCR conditions were 95 °C for 3 min followed by 40 cycles at 95 °C for 10 s and 60 °C for 30 s.
	R: CGGCATTGCTCGATATGGG	
Actin	F: GGCATGGAGTCTTCTGGAATCC	
	R: CCACCACTGAGCACTATGTTTC	

**Table 2 jof-09-00331-t002:** Two−factor ANOVA of the effect of *Epichloë* endophytic fungi on Na^+^, K^+^ content and K^+^/Na^+^ of *H. bogdanii* shoots and roots under alkali stress.

			Na*^+^*		K*^+^*		K*^+^*/Na*^+^*	
	deal	df	*F*	*p*	*F*	*p*	*F*	*p*
Shoot	E	1	215.727	<0.001	424.706	<0.001	1043.191	<0.001
	S	2	4705.524	<0.001	832.995	<0.001	2069.798	<0.001
	E × S	2	234.500	<0.001	252.975	<0.001	940.424	<0.001
Root	E	1	46.505	<0.001	2.612	<0.001	41.704	<0.001
	S	2	6256.824	<0.001	987.561	0.123	6249.812	<0.001
	E × S	2	28.093	<0.010	57.777	<0.001	101.181	<0.001

S = salt, E = endophytic fungus, E × S is the interaction between the two.

**Table 3 jof-09-00331-t003:** Two−factor ANOVA of the effect of *Epichloë* endophytic fungi on Cl^−^, SO_4_^2−^and NO_3_^−^contents in *H. bogdanii* under alkali stress.

			Cl^−^		SO_4_^2−^		NO_3_^−^	
	deal	df	*F*	*p*	*F*	*p*	*F*	*p*
Shoot	E	1	536.566	<0.001	0.013	0.910	495.057	<0.001
	S	2	1741.801	<0.001	183.331	<0.001	1115.885	<0.001
	E × S	2	568.100	<0.001	1.190	0.338	1610.349	<0.001
Root	E	1	41.525	<0.001	0.147	0.708	0.908	0.36
	S	2	4.145	0.033	189.965	<0.001	0.506	0.615
	E × S	2	10.504	<0.001	281.279	<0.001	0.236	0.793

S = salt, E = endophytic fungus, E × S is the interaction between the two.

## Data Availability

The datasets used and analyzed during the current study are available from the corresponding author upon reasonable request.

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
