# Peer review of "Endophytic Fungi Regulate HbNHX1 Expression and Ion Balance in Hordeum bogdanii under Alkaline Stress"

_jof, 2023, doi:10.3390/jof9030331_

Round 1

Reviewer 1 Report

This manuscript did not have sufficient attention to detail and did not follow instructions to authors for this journal. It was not well written and will need editing for grammar. Conclusions should not be included in the Results section. The Figure and Table legends need to be rewritten or written. The Methods do not provide numbers of plants or reps.

Abstract - remove (1) Background, (2) Methods, (3) Results, (4) Concluisons

Line 61 - capitalize "Plants"

Line 105 - change to "To avoid alkalai shock..."

Line 159 - "indicates ......" is a conclusion. Delete.

Line 160 - "Reaching...." is a sentence fragment. Reword.

Line 167-168 - This is a conclusion. Delete.

Line 171 - This is a conclusion. Delete.

Line 172 and 347 - change "regionalization" to "localization"

Line 176 - delete "indicating ......." which is a conclusion.

Table 2 legend - change to S=salt, SXE  to match table

Table 3 - add legend

Figure 3 - change to M=Marker

Figure 4 - add legend

Figures and Tables should be moved close to first mention in text.

Methods - How many plants were tested? Were there multiple pots of plants of each type? Were there reps? How were they done? Need to include the N for individual plants tested.

Author Response

Dear reviewer
Thank you for your letter and your comments concerning our manuscript entitled “Endophytic Fungi Regulate HbNHX1 Expression and Ion Balance in Hordeum bogdanii under Alkaline Stress ” (ID:2232504). Those comments are valuable and very helpful. We have read through comments carefully and have made corrections. Based on the instructions provided in your letter, we uploaded the file of the revised manuscript. Revisions in the text are shown using red highlight for additions, and strikethrough font for deletions.Our responses to reviewer comments are as follows:
Point 1: Abstract - remove (1) Background, (2) Methods, (3) Results, (4) Concluisons

Response 1: We are grateful for the suggestion. We have deleted the Background, Methods, Results and Concluisons in the summary.

Point 2: Line 61 - capitalize "Plants"

Response 2: We are grateful for the suggestion. We have replaced "plants" in line 61 with "Plants".

Point 3: Line 105 - change to "To avoid alkali shock..."
Response 3: We apologize for the language problems in the original manuscript. We have replaced "Avoid alkali shock" to "To avoid alkali shock".
Point 4: Line 159 - "indicates ......" is a conclusion. Delete.
Response 4: We agree with the comment and delete the conclusion
Point 5: Line 160 - "Reaching...." is a sentence fragment. Reword.
Response 5: We are grateful for the suggestion. As suggested by the reviewer, we have reword:  With an increase in alkaline concentration, the K+ content decreased continuously, but it was higher in E+ than that in E- (Figure 1(A, B)), and the content of K+ at 50 mmol·L-1 E+ is significantly higher than that of E-.

Point 6: Line 167-168 - This is a conclusion. Delete.
Response 6: We agree with the comment and delete the conclusion.

Point 7: Line 171 - This is a conclusion. Delete.
Response 7: We agree to comment on and delete the conclusions and rewrite the results.

Point 8: Line 172 and 347 - change "regionalization" to "localization"

Response 8: We agree with the comment. We have replaced "regionalization" to "localization".

Point 9: Line 176 - delete "indicating " 

Response 9: We agree with the comment and delete the conclusion.

Point 10: Table 2 legend - change to S=salt, SXE to match table

Response 10: We agree with the comment. We have modified it according to the way you instructed. S =salt, E = endophytic fungus, E × S is the interaction between the two.

Point 11: Table 3 and figure 4 - add legend

Response 11: We are grateful for the suggestion. We have added the legend to Table 3 and figure 4.

Point 12: Figure 3 - change to M=Marker

Response 12: We agree with the comment. We have modified it according to the way you instructed.

Point 13: Figures and Tables should be moved close to first mention in text.

Response 13: We agree with the comment. We have modified it according to the way you instructed.

Point 14: Methods - How many plants were tested? Were there multiple pots of plants of each type? Were there reps? How were they done? Need to include the N for individual plants tested.

Response 14: E+ and E- plump seeds were germinated in water until they grow needles and then transplanted into 4 L hydroponic boxes, 19 holes per pot, spaced at 1 seedling per hole for a total of 19. After 21 day of alkaline stress treatment, four plants were randomly selected from the hydroponics box for ion content determination.

We would love to thank you for allowing us to resubmit a revised copy of the manuscript and we highly appreciate your time and consideration.

Sincerely,

Long feng
Key Laboratory of the Corps for the Conservation and Utilization of Biological Resources in the Tarim Basin/College of Life Sciences and Technology

Tarim University, Alar, Xinjiang Province, 843300, China

Reviewer 2 Report

The manuscript “Endophytic Fungi Regulate HbNHX1 Expression and Ion Balance in Hordeum bogdanii under Alkaline Stress” represents an original scientific paper. The authors found that Epichloë endophytic fungi promoted the up regulation of HbNHX1 in the host H. bogdanii under alkaline environment, and participated in the regulation of anion and cation transport in the host plant. These results provide a scientific basis for the breeding of new plant varieties and improving the stress tolerance of new varieties under alkaline environment.

The title of the manuscript is informative and clearly reflects its contents. It definitely corresponds to the point of carried investigation and to the subject of the journal. The introduction summarizes relevant researches. Used methodology is appropriate. Results are clearly presented and adequately discussed.

Overall, this manuscript is written exceptionally well; clear and easy to understand.

Minor suggestions for improvement:

In some parts of article italics are not used for Latin names of plants and fungi.

p. 5, Line 225: “AB09137.1” is an erroneous GenBank number. Please correct it. This sequence isn't in Figure 5 (p. 9), but AB089197.1 (Hordeum) is in the tree, so I assume it's a typo.

The abbreviations below Table 2 do not match to those used in the table; replace "A" with "S", and "A × E" with "S × E".

Some names of authors cited in the text are incorrect:

p. 2, Line 59: Replace “Martín et al. [12]” with “Pérez-Martín et al. [12]”

p. 9, Line 294: Replace “Meiling [37]” with “Song [37]”

Author Response

Dear reviewer
Thank you for your letter and the your comments concerning our manuscript entitled “Endophytic Fungi Regulate HbNHX1 Expression and Ion Balance in Hordeum bogdanii under Alkaline Stress ” (ID:2232504). Those comments are valuable and very helpful. We have read through comments carefully and have made corrections. Based on the instructions provided in your letter, we uploaded the file of the revised manuscript. Revisions in the text are shown using red highlight for additions, and strikethrough font for deletions. Our responses to reviewer comments are as follows:
Point 1: In some parts of article italics are not used for Latin names of plants and fungi.

Response 1: We appreciate your advice. We reviewed other parts of the article and made changes.

Point 2: p. 5, Line 225: “AB09137.1” is an erroneous GenBank number. Please correct it. This sequence isn't in Figure 5 (p. 9), but AB089197.1 (Hordeum) is in the tree, so I assume it's a typo.

Response 2: We apologize for the erroneous in the original manuscript. We have corrected the wrong gene serial number in the paper, please check the changed content in the revised version, thank you again for the problem you found.

Point 3:  The abbreviations below Table 2 do not match to those used in the table; replace "A" with "S", and "A×E" with "S×E".

Response 3: We apologize for the erroneous in the original manuscript. We have change to S =salt, E = endophytic fungus, E × S is the interaction between the two.
Point 4:  Some names of authors cited in the text are incorrect:

  1. 2, Line 59: Replace “Martín et al.[12]” with “Pérez-Martín et al.[12]”
  2. 9, Line 294:Replace “Meiling[37]” with “Song [37]”
    Response 4: We thank you for finding the error in the manuscript and apologize for it. We have modified it according to the correct format you mentioned.

We would love to thank you for allowing us to resubmit a revised copy of the manuscript and we highly appreciate your time and consideration.

Sincerely.

Long feng

Key Laboratory of the Corps for the Conservation and Utilization of Biological Resources in the Tarim Basin/College of Life Sciences and Technology

Tarim University, Alar, Xinjiang Province, 843300, China

Round 2

Reviewer 1 Report

The authors fixed many of the concerns in the revision. There are still a few points that were not adequately addressed.

Figure 3 legend was not fixed as requested.

Line 171 is a sentence fragment.

Line 346 - change to mycorrhizae and this should not be in italics since it is not a genus species name.

Figure 4 legend is incomplete. What type of analyses are these? Need more explanation in the figure legend in addition to the text. The methods for this were not in the methods section. They should be listed there instead of in the results section.

Figure 5 legend is incomplete. What type of analysis was run? Is this from neighbor joining or parsimony or ? Once again the methods for this should be in the methods section.

Lines 96 and 109 - Still unclear on the numbers. I understand that there were 2 boxes with 19 plants each. Four plants were removed from each box for further testing after alkaline stress. What of this was replicated or repeated?

Author Response

Dear reviewer

Thank you for your letter and your comments concerning our manuscript entitled “Endophytic Fungi Regulate HbNHX1 Expression and Ion Balance in Hordeum bogdanii under Alkaline Stress ” (ID:2232504). Those comments are valuable and very helpful. We have read through comments carefully and have made corrections. Based on the instructions provided in your letter, we uploaded the file of the revised manuscript. Revisions in the text are shown using red highlight for additions. Our responses to reviewer comments are as follows:

Point 1: Figure 3 legend was not fixed as requested.

Response 1: We apologize for the spelling problems in the revision. We have corrected this problem.

Point 2: Line 171 is a sentence fragment.

Response 2: We are grateful for the suggestion. We have perfected the sentence meaning of line 171.

Point 3: Line 346 - change to mycorrhizae and this should not be in italics since it is not a genus species name.
Response 3: We agree to make a comment and change Piriformospora indica to mycorrhizae.
Point 4: Figure 4 legend is incomplete. What type of analyses are these? Need more explanation in the figure legend in addition to the text. The methods for this were not in the methods section. They should be listed there instead of in the results section.
Response 4: We agree with your comments, we have improved the legend in Figure 4 and put the analysis method in the data analysis of the material method, and we have recounted it in the result section.

Point 5: Figure 5 legend is incomplete. What type of analysis was run? Is this from neighbor joining or parsimony or ? Once again the methods for this should be in the methods section.

Response 5: We agree with your comments, we have improved the legend in Figure 5, and put the analytical method used in the data analysis of the materials method, which is recounted in the results section.
Point 6: Lines 96 and 109 - Still unclear on the numbers. I understand that there were 2 boxes with 19 plants each. Four plants were removed from each box for further testing after alkaline stress. What of this was replicated or repeated?

Response 6: Thank you for your comments on the deficiencies in the revised manuscript. Our experiment used an experiment had a randomized complete block design with four replicates (hydroponic box) of E+ and E- seedlings. We modified this part.

We would love to thank you for allowing us to resubmit a revised copy of the manuscript and we highly appreciate your time and consideration.

Sincerely,
Long feng
Key Laboratory of the Corps for the Conservation and Utilization of Biological Resources in the Tarim Basin/College of Life Sciences and Technology

Tarim University, Alar, Xinjiang Province, 843300, China
